# The origin of the Moon's Earth-like tungsten isotopic composition from dynamical and geochemical modeling

Rebecca A. Fischer [1,3✉], Nicholas G. Zube [2,3] & Francis Nimmo [2]

The Earth and Moon have identical or very similar isotopic compositions for many elements, including tungsten. However, canonical models of the Moon-forming impact predict that the Moon should be made mostly of material from the impactor, Theia. Here we evaluate the probability of the Moon inheriting its Earth-like tungsten isotopes from Theia in the canonical giant impact scenario, using 242 N-body models of planetary accretion and tracking tungsten isotopic evolution, and find that this probability is <1.6–4.7%. Mixing in up to 30% terrestrial materials increases this probability, but it remains <10%. Achieving similarity in stable isotopes is also a low-probability outcome, and is controlled by different mechanisms than tungsten. The Moon's stable isotopes and tungsten isotopic composition are anticorrelated due to redox effects, lowering the joint probability to significantly less than 0.08–0.4%. We therefore conclude that alternate explanations for the Moon's isotopic composition are likely more plausible.

[1] Department of Earth and Planetary Sciences, Harvard University, Cambridge, USA. [2] Department of Earth and Planetary Sciences, University of California Santa Cruz, Santa Cruz, USA. [3] These authors contributed equally: Rebecca A. Fischer, Nicholas G. Zube. ✉email: rebeccafischer@g.harvard.edu

The Earth and Moon have similar or identical isotopic compositions for many elements, including O (e.g., see refs. [1,2]), Ti[3], and Cr (e.g., see ref. [4]). They have slightly different W isotopic compositions, but this difference is approximately equal to the effects of disproportionate chondritic late accretion, so that their pre-late veneer W isotopic compositions were likely the same[5,6]. These observations have inspired many theories of lunar formation, including different dynamical regimes for the Moon-forming impact that create more mixing[7–10], post-impact Earth–Moon isotopic equilibration via vaporized silicate[11,12], and Earth and Theia coincidentally having the same isotopic composition (e.g., see ref. [13]).

Several previous studies have evaluated the likelihood of the Moon's Earth-like oxygen isotopic composition being inherited from Theia, finding probabilities ranging from <2% (see ref. [11]) to 5–8% (see ref. [14]) to 20–40% (see ref. [15]). It has also been shown that this Theia-inheritance mechanism can reproduce the Moon's Earth-like Ca, Ti, and Cr isotopic compositions, but the probabilities of doing so were not evaluated[16].

The Moon's tungsten isotopic composition provides a fundamentally different and more restrictive constraint than these stable isotopes (e.g., see ref. [17]). In the early Solar System, the now-extinct $^{182}$Hf decayed to $^{182}$W with a half-life of 9 Ma. Hf is lithophile, whereas W is moderately siderophile, so they are fractionated by core formation. A body's mantle W isotopic composition depends on the timescales of accretion and differentiation and on the physical mechanisms of core formation (pressure, temperature, oxygen fugacity, degree of metal–silicate equilibration) (e.g., see refs. [18–21]), not on source material.

Previous studies of the Moon's W isotopic composition have used inverse[13] or forward[22] models to find parameters that can explain the observed composition, demonstrating that it is possible for the Moon to inherit an Earth-like W isotopic composition from Theia but without constraining its probability. One study used a forward model of Hf–W isotopic evolution based on a small number of N-body simulations, showing that it is not possible to form a Theia analog with both a Moon-like W isotopic composition and a Moon-like Hf/W ratio[19]. The N-body simulations used in that study[23] are similar to the Circular Jupiter and Saturn (CJS) and Eccentric Jupiter and Saturn (EJS) simulations used here[24], with the main differences being a higher resolution (more initial bodies) in the simulations used here and a much larger number of simulations (100 CJS/EJS in this study versus 8 in ref. [19]) to allow for more quantitative assessment of probabilities. A Monte Carlo model showed that producing an Earth and Moon with the same W isotopic composition by this mechanism has a probability of <1%, if most (>80%) lunar material derived from Theia[17].

Here, we take a different approach, using detailed forward models of accretion and core formation based on a large number of dynamical simulations to better constrain the probability of an Earth-like lunar W isotopic composition inherited from Theia. We also consider the formation of the Moon from an Earth–Theia mixture, but do not directly quantify the probabilities of large extents of mixing or post-impact equilibration since these hypotheses are sensitive to different processes (e.g., dynamics of the Moon-forming impact instead of core formation). We find that the probability of the Moon inheriting an Earth-like tungsten isotopic composition from Theia is <1.6–4.7%; forming the Moon from up to 30% terrestrial materials increases this probability, but it remains <10%. Reproducing the Moon's Earth-like stable isotopic composition is anticorrelated with tungsten due to redox effects, decreasing the joint probability to <0.08–0.4% and suggesting that other explanations for the Moon's isotopic composition may be more likely.

## Results and discussion

**Terrestrial and lunar $^{182}$W anomalies.** Our models are based on a large suite of N-body simulations of terrestrial planet accretion in the inner Solar System: 50 CJS (see ref. [24]), 50 EJS (see ref. [24]), and 142 Grand Tack (see ref. [25]) simulations. Each simulation began with ~80–100 larger planetary embryos and ~2000 smaller planetesimals, whose orbital evolutions were tracked for 150–200 Ma until they accreted into planets (see refs. [24,25]) (see "Methods" section). The simulations also included the Sun, Jupiter, and Saturn. The three accretion scenarios considered here differ primarily in the orbits and migration histories/timescales of the gas giant planets, and their resulting effects on the extent and timescales of radial mixing in the disk (e.g., see ref. [26]).

We tracked the isotopic evolution of Earth and Theia analogs formed in these simulations (see "Methods" section) by coupling the simulations to models of protracted core formation, calculating their Hf–W isotopic signatures as tungsten partitions between metal and silicate with each impact. We included the effects of partial metal–silicate equilibration and an initial oxygen fugacity gradient in the disk (see "Methods" section). The radiogenic $^{182}$W excess in a body's mantle is expressed as a $^{182}$W anomaly, $\varepsilon_{182W} = \left[\frac{(X^{182W}/X^{184W})}{(X^{182W}/X^{184W})_{CHUR}} - 1\right] \times 10^4$, where $X^{182W}$ and $X^{184W}$ are mole fractions of $^{182}$W and $^{184}$W, respectively, and CHUR is the chondritic uniform reservoir. The Earth and Moon both have inferred pre-late veneer $\varepsilon_{182W} = 2.2$ (see ref. [5]), with a measurement uncertainty of approximately ±0.05 $\varepsilon$ units on lunar samples (see refs. [5,6]) and an uncertainty of ±0.15 $\varepsilon$ units based on the calculated pre-late veneer composition of the Earth (see ref. [5]). $\varepsilon_{182W}$ depends on the abundance of radiogenic $^{182}$Hf (and thus Hf in general), as well as the abundance of a stable isotope of W. Therefore it is sensitive to the relative concentrations of Hf and W, quantified as $f^{Hf/W} = \frac{(X^{180Hf}/X^{184W})}{(X^{180Hf}/X^{184W})_{CHUR}} - 1$.

Figure 1 shows the evolution of mass, mantle $WO_3$ (the portion of a body's W budget hosted in silicates), and $\varepsilon_{182W}$ in two example Earth–Theia pairs. The Earth's $WO_3$ generally increases as it grows, because W becomes less siderophile at greater depths. If Theia originates in the inner circumstellar disk with a reduced composition, its $WO_3$ starts low and remains low due to the low pressures and temperatures of its interior. If it starts in the outer disk with an oxidized composition, its $WO_3$ is initially higher. Seventy-three percent of Theia analogs in CJS/EJS simulations and 23% in Grand Tack simulations have mass-weighted semimajor axes (MWSMA) of 1.5 AU or greater, giving them more oxidized compositions. $\varepsilon_{182W}$ in Earth and sometimes Theia is very high at early times (e.g., see ref. [21]). This is because reduced conditions and low pressures and temperatures cause W to be more strongly siderophile (a high W metal–silicate partition coefficient, $D_W = \frac{X^W}{X^{WO_3}}$), resulting in low mantle $WO_3$ and a higher proportion of radiogenic $^{182}$W in the mantle.

Theia often has a higher $\varepsilon_{182W}$ than Earth throughout much of its evolution. Theia typically finishes its accretion earlier than Earth, causing it to evolve to a higher $\varepsilon_{182W}$[19]. Theia is also smaller than Earth, and the lower pressures and temperatures make W more siderophile, which also increases $\varepsilon_{182W}$. Oxidized Theia analogs from the outer disk have lower $\varepsilon_{182W}$, since W is less siderophile under oxidizing conditions.

The Earth and Theia were tracked until their collision, then a Moon was formed from Theia material with single-stage core formation and no post-impact mixing or equilibration (see "Methods" section). As an endmember case, we begin by considering the formation of the Moon from entirely Theia materials, then later discuss the effects of mixing Earth and Theia materials to form the Moon. Figure 2 shows terrestrial and lunar $\varepsilon_{182W}$, evolved to the present, for various degrees of metal

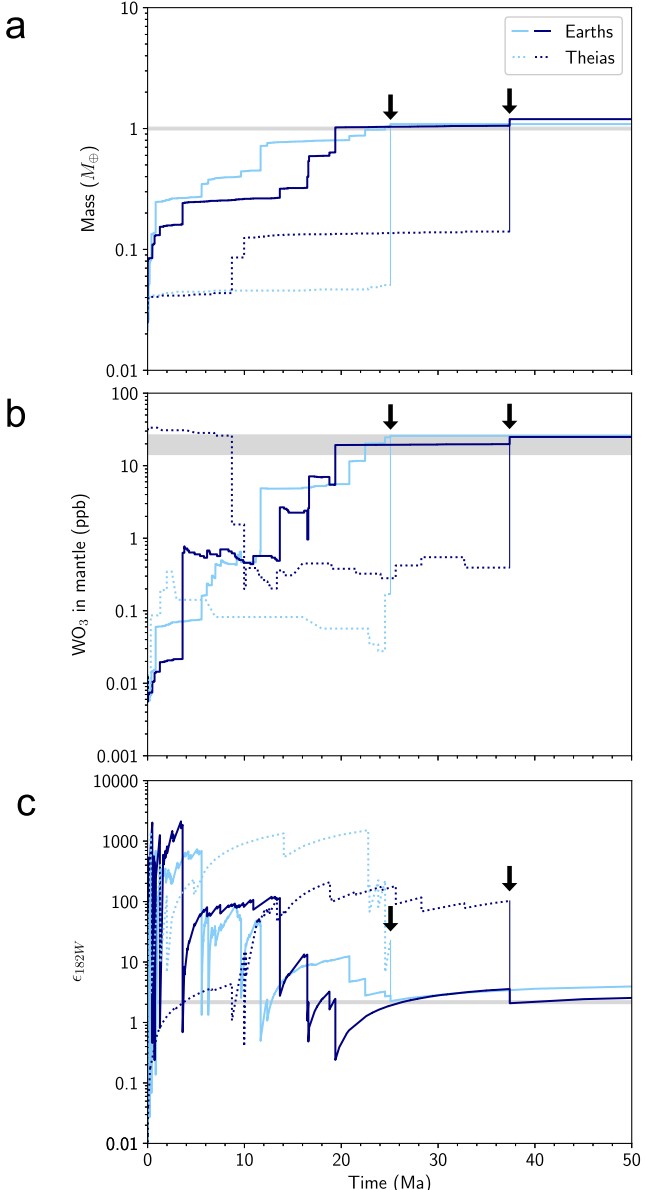

**Fig. 1 Evolution of mantle W in Earth and Theia analogs.** As Earth and Theia accrete, their masses increase in a stepwise fashion (**a**). Their mantle tungsten abundances ($WO_3$ in mantle) also generally increase, depending on the evolution of their oxidation states (**b**). Their mantle $\varepsilon_{182W}$ increase between impacts as $^{182}$Hf decays into $^{182}$W, faster at earlier times and slower after more half-lives of $^{182}$Hf (~9 Ma) have passed. With each impact, metal–silicate equilibration draws down the $^{182}$W in the mantle, decreasing $\varepsilon_{182W}$ (**c**). $\varepsilon_{182W}$ can reach very high values at early times, because W is more strongly siderophile at low pressure–temperature–oxygen fugacity conditions. Two example Earth-Theia pairs are shown (light and dark blue) from Eccentric Jupiter and Saturn (EJS) simulations. Solid lines: Earth analogs. Dotted lines: Theia analogs. Gray shaded regions indicate Earth values[5,43]. Thin vertical solid lines with black arrows represent the Moon-forming impacts.

equilibration $k$, assuming the Moon formed entirely from Theia materials. For $k = 0.4$, CJS/EJS Earth analogs match the observed $\varepsilon_{182W}$ on average[21], but many Moon analogs have anomalies that are too high. For a lunar $D_W = 30$, 75% of CJS/EJS Moon analogs have $\varepsilon_{182W}$ that are higher than the observed value (median 3.9), compared to 48% of Earth analogs (median 2.1). For $k = 0.8$, Grand Tack Earth analogs match the observed $\varepsilon_{182W}$ on

average[27], but again many Moon analogs have high anomalies. For a lunar $D_W = 30$, 85% of Grand Tack Moon analogs have $\varepsilon_{182W}$ that are higher than the observed value (median 6.7), compared to 54% of Earth analogs (median 2.3).

**Probability of an Earth-like lunar $\varepsilon_{182W}$.** On average, modeled lunar $\varepsilon_{182W}$ are higher than the measured value when the Moon analog is formed from Theia material (Fig. 2)[17], but some Moon analogs have lower anomalies. Because planetary accretion is a highly stochastic process, we now consider only model runs that produce an Earth-like pre-late veneer $^{182}$W anomaly of $2.2 \pm 0.15$ in our Earth analogs[5], and evaluate how many of these cases result in a Moon analog with a similar anomaly. We use values of $k$ ranging from 0.1 to 1, and combine results from different $k$ to achieve better statistics.

Figure 3 shows the probability of the Earth and Moon analogs having the same $\varepsilon_{182W}$ within a given tolerance, given that the Earth analog has an Earth-like anomaly. With a tolerance of $\pm 0.15$ $\varepsilon$ units based on the uncertainty in the Earth's pre-late veneer composition[5], and forming the Moon entirely from Theia materials (Fig. 3a), there is a <1.6–4.7% chance of the Moon analog coincidentally having an Earth-like $\varepsilon_{182W}$ via this mechanism. Using a smaller tolerance than $\pm 0.15$, more similar to measurement precision (e.g., see refs. [5,6]), would result in an even lower probability of forming a Moon with an Earth-like $\varepsilon_{182W}$. This probability applies to CJS, EJS, and Grand Tack scenarios, and to different values of $k$ and lunar $D_W$ (values of $D_W$ in the range 30–250 were not found to have significantly different probabilities of an Earth–Moon match). Note that at larger Earth–Moon tungsten anomaly differences, the Grand Tack scenario generally produces lower cumulative probabilities, since its planets form faster, resulting in larger $\varepsilon_{182W}$[27]. A probability of <1.6–4.7% is qualitatively consistent with earlier work showing that it is possible to produce an Earth-like $\varepsilon_{182W}$ in the Moon in some instances[13,19,22,28]. Our findings agree well with an earlier study suggesting that this is a low probability event, with a likelihood on the order of 1% (see ref. [17]), but here we have quantified this probability using dynamically-consistent accretion and core formation models, with provenance and mass evolution constrained by N-body simulations.

This probability can be increased by considering a Moon formed from a mixture of Earth and Theia. For example, for CJS/EJS simulations and a lunar $D_W = 30$, Moons formed entirely of Theia materials have a 1.9% chance of matching the Earth's $\varepsilon_{182W}$ within $\pm 0.15$ $\varepsilon$ units. Forming the Moon out of 90% Theia + 10% Earth increases this probability to 3.8%, while using 70–80% Theia + 20–30% Earth increases the probability to 9.4%, and using 50% Theia + 50% Earth results in a 17.0% probability of a match (Fig. 3b). Similar results were found using $D_W = 150$: for example, a Moon formed from 70% Theia + 30% Earth has a 7.5% probability of matching the Earth's $\varepsilon_{182W}$ within $\pm 0.15$ $\varepsilon$ units. These probabilities are slightly higher than those reported in a previous study based on a Monte Carlo model[17], which found an Earth–Moon $\varepsilon_{182W}$ match in <1% of cases when >80% of lunar material derived from Theia, and in <5% of cases when <20% of the Moon derived from Theia, though the results of these two studies are in qualitative agreement that an Earth–Moon match is a low probability event.

**Implications for lunar formation scenarios.** It is possible to form a Moon with an Earth-like $\varepsilon_{182W}$ by inheriting it primarily from Theia, but the probability is very low. The Hf–W isotopic system provides a different constraint than stable isotopes, reflecting the timing and mechanisms of core formation rather than source regions. Producing an Earth-like oxygen isotopic

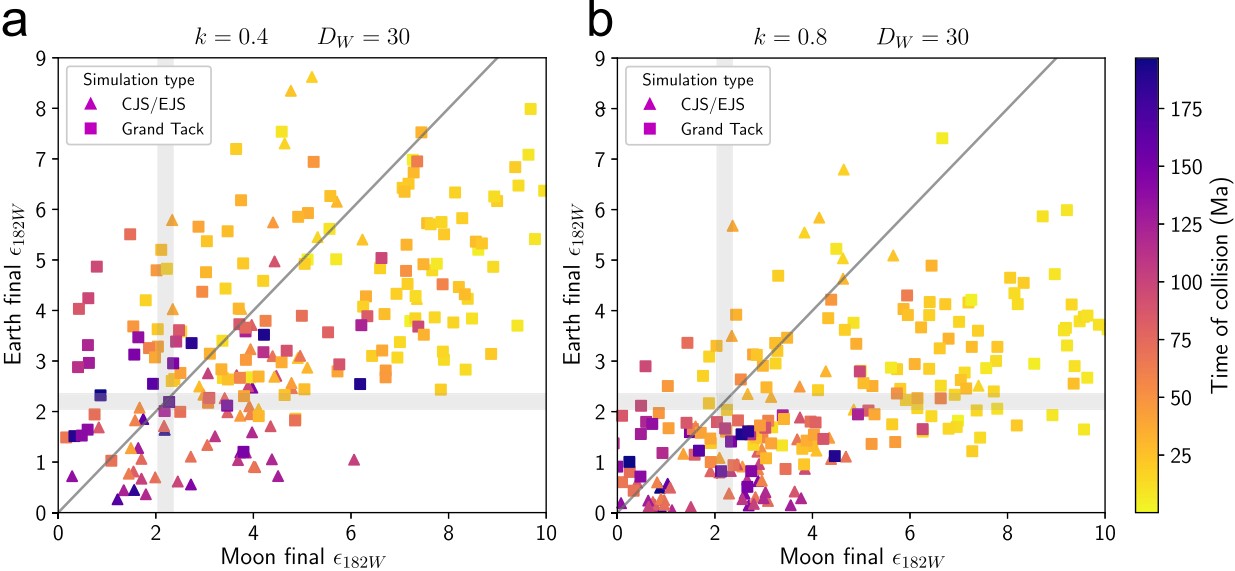

**Fig. 2 Final $\varepsilon_{182W}$ of Earth–Moon pairs.** The 1:1 line represents an identical $^{182}$W anomaly in the Earth and Moon as observed, a low-probability event. Gray shaded boxes: actual pre-late veneer $\varepsilon_{182W}$ in the Earth and Moon[5]. Symbol shape indicates accretion scenario; symbol color indicates the time of the Moon-forming collision (expressed as time since Solar System formation). Results are shown for a lunar W metal–silicate partition coefficient $D_W = 30$ and whole mantle equilibration. In **a**, the fraction of equilibrating metal was $k = 0.4$, which provides the best match to Earth's $\varepsilon_{182W}$ on average in Circular Jupiter and Saturn (CJS) and Eccentric Jupiter and Saturn (EJS) scenarios;[19,21] in **b**, it was $k = 0.8$, which provides the best match to Earth's $\varepsilon_{182W}$ on average in the Grand Tack scenario[27]. Lunar $\varepsilon_{182W}$ was calculated assuming that the Moon formed entirely from Theia materials, with no terrestrial component. While CJS/EJS Earth analogs in **a** and Grand Tack Earth analogs in **b** match the observed $\varepsilon_{182W}$ on average, 75 and 85% of Moon analogs, respectively, have anomalies that are too high.

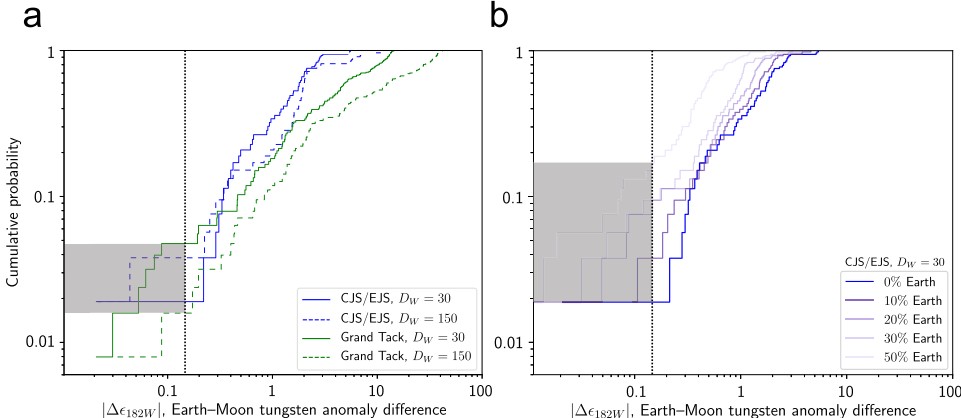

**Fig. 3 The probability of an Earth-like lunar $\varepsilon_{182W}$.** The cumulative probability of a range of Earth–Moon $\varepsilon_{182W}$ differences is shown, given an Earth-like terrestrial $\varepsilon_{182W}$ of 2.2 ± 0.15 (see ref. [5]). Vertical dotted lines indicate the maximum Earth–Moon $\varepsilon_{182W}$ difference allowed based on calculations of the Earth's pre-late veneer composition, ±0.15 $\varepsilon$ units (see ref. [5]), and the gray shaded rectangles indicate the corresponding maximum probability of forming an Earth–Moon pair with an $\varepsilon_{182W}$ difference no larger than this. Results for various fractions of equilibrating metal $k$ (in the range 0.1–1) are combined. **a** Results for four combinations of accretion scenarios and lunar W metal–silicate partition coefficient $D_W$, with the Moon made entirely of Theia materials (no terrestrial component). Blue lines: Circular Jupiter and Saturn (CJS) and Eccentric Jupiter and Saturn (EJS) simulations, combined. Green lines: Grand Tack simulations. Solid lines: $D_W = 30$. Dashed lines: $D_W = 150$. With the Moon made entirely from Theia materials, <1.6–4.7% of Earth–Moon pairs have $\varepsilon_{182W}$ differences of <0.15 $\varepsilon$ units. **b** The effect of mixing Earth and Theia materials to form the Moon, with 0–50% of the mass of the Moon coming from the Earth (indicated by color). Results are shown for CJS/EJS simulations and $D_W = 30$. In this case, the probability of an Earth–Moon pair having an $\varepsilon_{182W}$ difference of <0.15 $\varepsilon$ units ranges from 1.9% (0% Earth materials) to 9.4% (20–30% Earth materials) to 17.0% (50% Earth materials). These probabilities are for W isotopes alone, and may be significantly lower when also considering stable isotopes.

composition in the Moon (e.g., see refs. [1,2]. cf.[29]) would have also been a low-probability event (e.g., see refs. [11,14,15]). If the Moon's isotopic composition was indeed inherited from Theia, both of these low probability events would have to have occurred. With a probability of 5–8% for oxygen (see ref. [14]) and <1.6–4.7% for tungsten, the likelihood of the Moon inheriting Earth-like O and

W isotopic signatures simultaneously is only <0.08–0.4%. This probability may be lower by a factor of >3–4 (see ref. [11]) or higher by a factor of ~4–5 (see ref. [15]) if different estimates for the probability of an Earth-like lunar O isotope signature are used.

This analysis assumes that inheriting Earth-like O and W isotopic compositions are independent events, but in actuality

they are anti-correlated, so the true probability is even lower than this. Producing a Moon with a Moon-like $f^{Hf/W}$ requires the accretion of a lot of oxidized material from the outer disk, otherwise tungsten is too siderophile at the low pressures and temperatures of the interiors of Theia and the Moon. Moon analogs with a Moon-like $f^{Hf/W}$ and $\varepsilon_{182W}$ have MWSMA much more similar to Mars than to Earth (Supplementary Fig. 1). Mars and Earth have different O isotopic compositions (e.g., see ref. [30]). If Theia originated from a Mars-like MWSMA and the Moon derived primarily from Theia, the Moon would be expected to have a stable isotopic composition more similar to Mars than to Earth, contrary to observations, though there is considerable uncertainty in the initial stable isotope distribution in the protoplanetary disk (e.g., see ref. [14]). Moon analogs with an Earth-like MWSMA have a very high $f^{Hf/W}$ of ~100 and $\varepsilon_{182W}$ values that are significantly higher than the Moon's on average (Supplementary Fig. 1).

A previous study concluded that Theia had to have $f^{Hf/W}$ of ~8–30 and $\varepsilon_{182W}$ of ~3.5–7 to produce a Moon with the observed values, based on Monte Carlo modeling[13]. Our CJS/EJS simulations produce 20 Theia analogs with MWSMA <1.5 AU (similar to the Earth, to match the stable isotope constraints). Of these Theia analogs, 0/20 have $f^{Hf/W}$ = 8–30 (median 1100) and 3/20 have $\varepsilon_{182W}$ = 3.5–7 (median 34). Similarly, our Grand Tack simulations produce 186 Theia analogs with MWSMA <1.5 AU, of which 2/186 have $f^{Hf/W}$ = 8–30 (median 1700) and 14/186 have $\varepsilon_{182W}$ = 3.5–7 (median 25), due to the low pressures and temperatures of their interiors, reduced conditions in the inner disk, and rapid accretion. While in principle these Theia $f^{Hf/W}$ and $\varepsilon_{182W}$ could produce a Moon having an Earth-like isotopic composition, our dynamical models show that such a parameter combination in the inner disk is very unlikely.

Because of the anti-correlated constraints, the probability of producing a Moon with an Earth-like isotopic composition inherited from Theia is actually significantly lower than the <1.6–4.7% chance of matching $\varepsilon_{182W}$ alone, or the <0.08–0.4% if tungsten and oxygen isotopes were independent. As a result, we consider the Theia inheritance model to be difficult to reconcile with isotopic observations. Alternative models such as enhanced Earth–Theia mixing (e.g., see refs. [7–10]) or post-impact equilibration (e.g., see refs. [11,12]) are likely to have a higher probability of reproducing the observed isotopic compositions. For example, high energy giant impacts that can potentially allow post-impact isotopic equilibration were common in the end-stages of accretion, with ~85% of bodies experiencing at least one late giant impact with a modified specific impact energy of >2 × $10^6$ J/kg (see ref. [12]). Achieving a high degree of Earth–Theia mixing when the two bodies are similar sizes requires impact velocities that are common in accretion simulations, with ~20% of planets experiencing an impact with a mass ratio of ≥0.4 (see ref. [7]). Future studies should focus on further quantifying the probabilities of these scenarios using dynamical and geochemical constraints.

## Methods

***N-body accretion simulations.*** The 50 CJS and 50 EJS simulations[24] were run using the MERCURY code[31]. The CJS case had Jupiter and Saturn on non-eccentric orbits as predicted by the Nice Model (e.g., see ref. [32]). In the EJS scenario, Jupiter and Saturn were placed on their modern-day orbits. The 142 Grand Tack simulations[25] were run using the Symba code[33]. The Grand Tack scenario involves an inward then outward migration of Jupiter and Saturn to truncate the inner disk[34]. These simulations started with ~100 planetary embryos and ~2000 planetesimals. Different simulations featured different embryo:planetesimal mass ratios (1:1, 2:1, 4:1, or 8:1) and different embryo masses (0.025, 0.05, or 0.08 $M_\oplus$). In all CJS, EJS, and Grand Tack simulations used here, all collisions were treated as inelastic mergers, neglecting the possibility of hit-and-run or erosive impacts (e.g.,

see ref. [35]). This is a reasonable approximation, as incomplete accretion has been shown to have only a small effect on the resulting $\varepsilon_{182W}$[36]. These simulations have been previously published, and further details may be found in those studies[24,25].

***W isotopic calculations.*** Geochemical calculations were performed using two different models of Hf–W isotopic evolution during core formation, a "full model" and a "fast model". The full model consisted of a detailed model of core formation[21,37] tracking 13 elements, including Hf and W. This model included a spatial gradient in initial oxygen fugacity in the disk, constrained to reproduce the FeO contents of the terrestrial and Martian mantles on average: 4 log units below the iron–wüstite buffer (IW–4) inside of 1.5 AU, and IW–1.5 outside of 1.5 AU (e.g., see refs. [37,38]). As planets accreted and differentiated, their oxygen fugacities were evolved self-consistently[39]. The metal–silicate partition coefficient of tungsten $D_W$ (as well as those of all other elements) varied with changing pressure, temperature, composition, and oxygen fugacity[40] in a planet's interior as it grew. The full model was primarily used in calculations based on CJS/EJS N-body accretion simulations[24]. The fast model was simpler, designed for more rapid exploration of parameter space[27]. This model tracked only Hf and W, and used a constant value for $D_W$, with a spatial gradient in initial $f^{Hf/W}$ in the disk[19] to reproduce the observed variation in $f^{Hf/W}$ between Earth and Mars. The fast model was primarily used in calculations based on Grand Tack N-body simulations[25].

The full model identified Earth analogs as the largest surviving planet with a mass of 0.67–1.5 $M_\oplus$ and a semimajor axis of 0.75–1.25 AU, and the fast model identified Earth analogs as any surviving planet with a mass of 0.5–2 $M_\oplus$ and a semimajor axis of 0.387–1.524 AU. In both models, Theia analogs were defined as the last body to collide with the Earth that contained at least one planetary embryo. Both models included variable degrees of metal equilibration $k$, used here with whole mantle equilibration. Results from CJS and EJS simulations were combined, due to their resulting isotopic signatures being statistically indistinguishable. Results from different values of $k$ were combined to obtain probabilities. The two models were benchmarked against each other using Grand Tack simulations[25], showing agreement in Earth's $\varepsilon_{182W}$ to within typically <0.4 $\varepsilon$ units (Supplementary Fig. 2). Both of these models have been previously applied to studying Earth analogs; further details of the models may be found in those studies[21,27].

The Hf–W isotopic evolution of Earth and Theia analogs was tracked until the time of the Moon-forming impact. A Moon analog was formed from a portion of Theia, having the mass of the Moon and a core that was 2% of its total mass. For most of the results shown here, no terrestrial material was incorporated into the Moon analog, as an end-member case. We also considered cases where the lunar mantle was derived from a mixture of Theia and Earth materials, with up to 50% Earth materials by mass. The Moon analog underwent an episode of single-stage core formation, using a constant $D_W$ in the range 30–250 (e.g., see refs. [13,17,19,41,42]). The Earth analog was assumed to accrete the Theia analog (Theia analogs were much larger than the Moon). In Grand Tack simulations, the Earth analog was also assumed to accrete the late veneer (defined as all planetesimals accreted by an Earth analog after the Moon-forming impact, regardless of mass), since the late veneer in these simulations usually contributes a substantial fraction of the Earth analog's mass. In CJS/EJS simulations, the small late veneer was not accreted by the Earth analog. Finally, the Earth and Moon analogs were isotopically evolved to the present for comparison with observations.

## Data availability
The data that support the findings of this study are available within the paper and its supplementary information files, or are available from the corresponding author upon reasonable request.

## Code availability
Custom computer codes used to generate the results reported in this paper are available from the corresponding author upon reasonable request.

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

## Acknowledgements
This work was funded by a NASA Emerging Worlds grant (NNX17AE27G) to R.A.F. and F.N. We are grateful to Seth Jacobson for providing output of Grand Tack N-body simulations and to Stephanie Brown for helpful discussions.

## Author contributions
R.A.F. and N.G.Z. developed numerical models of W isotopic evolution and performed calculations. All authors discussed and interpreted results. R.A.F. wrote the manuscript with help from N.G.Z. and F.N.

## Competing interests
The authors declare no competing interests.
