## [Peer Review File · Nature Communications]

REVIEWER COMMENTS

Reviewer #1 (Remarks to the Author):

I have read and reviewed the manuscript by Fischer, Zube, and Nimmo. This work uses N-body simulations of terrestrial planet formation combined with a geochemical model of core formation to track the evolution of tungsten anomalies within Earth and Theia analogues. These results are then used to show that under several different terrestrial planet formation scenarios, there is only a ~5% or less chance of replicating the Earth's and Moon's similar tungsten anomalies. This result is critical because tungsten anomaly is largely set by accretion timescale and core formation. Meanwhile, other isotopes in which the Earth and Moon are similar are set through the source region of the planetesimal building blocks of Earth and Theia. Because previous works have shown that replicating the Earth and Moon's other isotopic similarities is also ~5%, this means that replicating both tungsten and non-tungsten similarities is exceptionally improbable within the canonical giant impact hypothesis that builds the Moon primarily from Theia material. This implies that either greater mixing took place between the proto-Earth and Theia and/or alternative impact scenarios must be invoked to explain the Earth and Moon's isotopic similarities. The tension between the canonical giant impact hypothesis (in which the Moon is largely derived from Theia material) and the isotopic similarities between Earth and Moon has been the subject of many past papers. This current work is thus highly significant and represents a breakthrough in this topic. After some modest revision, I believe it is suitable for publication.

My limited comments are as follows:

- Much more detail needs to be presented in how the authors chose relevant simulated systems and planets for analysis. How are Earth and Theia analogs defined within N-body simulations? What are the mass and orbital element bounds of Earth analogs? Are Theia analogs the last major impactor on Earth analogs? If so, what is considered to be a major impactor? Moreover, what fraction systems that contain an Earth and Theia analog also have a similar broader architecture to the inner solar system? If this fraction is low, how does this affect the validity of the authors' results?
- In the manuscript, the authors consider Moons that are 100% derived from Theia. How much do the results change if the authors make a less stringent requirement? For instance, does replicating the Earth-Moon tungsten anomaly become substantially more plausible if we allow 10% of the Moon to be derived from the proto-Earth? Even more useful for follow-up work studying the giant impact details: how much proto-Earth contribution to the Moon is necessary to increase the replication probability to the tens of percent level?
- The EJS and CJS simulation results are grouped together throughout the paper. Are their results statistically indistinguishable? If not, why are they grouped this way?
- What is the rationale for choosing $D_w=30$ and $D_w=150$ for the analysis? Can better W anomalies be attained with other values of D_w ? If so, are they physically plausible? I would ask the same question for k .
- Page 11, lines 171-175: The authors discuss the number of Theia analogues with feeding zones similar to Earth in the CJS and EJS batches and proceed to break down their W statistics. The same should be done for the Grand Tack simulations and their Theia analogs.

This concludes my comments.

Reviewer #2 (Remarks to the Author):

Review of "The origin of the Moon's Earth-like tungsten isotopic composition from dynamical and geochemical modeling" by Rebecca A. Fischer, Nicholas G. Zube, and Francis Nimmo submitted to Nature Communications.

The study by Fischer et al. aims to shed light on the origin of the Moon by a giant impact, a long-standing debate in planetary science. The key conundrum in this debate is that the Earth and the Moon are isotopically very similar, whereas canonical giant impact scenarios predict a larger fraction of the impactor (Theia) in the Moon, and hence an isotopic difference. The authors aim to tackle this question by assessing the likelihood of producing similar tungsten (W) isotope composition for the Earth and the Moon. To this end, they use sophisticated N-body model simulations for planetary accretion and simultaneously track the W isotopic evolution. The authors find that the probability of producing similar W isotopic compositions in canonical giant impact models is very small (<5%), and even smaller (<1%) if the similar O isotopic compositions of the Earth and Moon are taken into account as well. These results confirm conclusions drawn in other recent work. However, the model calculations used by the authors are considerably more elaborate, allowing for a more rigorous and quantitative assessment of the likelihood of the canonical giant impact scenario. In addition, a novel aspect of this study is that it simultaneously assesses the likelihood of producing similar W and O isotopic compositions for the Earth and the Moon.

The model calculations are state-of-the-art, and as far as I can tell, of excellent quality. The results of these calculations provide new and important insights into the likelihood of lunar formation scenarios, and hence, more generally into the mechanisms of planetary accretion. All things considered, I believe that this is a very timely and important contribution.

However, I do have several comments and suggestions that should be addressed before the paper can be published. Although the paper generally is very well-written, I believe that some improvements could be made with respect to readability of the paper. Hence, several of my comments below are intended to clarify certain aspects that seemed unclear and/not well enough explained in the current version of the manuscript.

Comments and suggestions (Line by line, most important comments in bold):

Line 14-15: "...coincidentally similar isotopic compositions for the proto-Earth and Theia. Here we evaluate the probability of this third option..." From this I gather that the authors assess the likelihood of having indistinguishable isotopic compositions only in the context of the canonical giant impact scenario. This should be pointed out more explicitly. In addition, it would be good to clarify why the likelihood of alternative giant impact scenarios (e.g. those involving larger contributions of the Earth's mantle and/or post giant impact equilibration) is not assessed in this study.

Lines 41-42: " ^{182}Hf decays to ^{182}W with a half-life of 9 Ma." Yes, but this decay system was only effective early in Solar System history, when ^{182}Hf was alive. This should be clarified to avoid confusion.

Lines 51-53: It would be helpful for the reader to explain how the N-body simulations performed by Nimmo et al. 2010 are different from those performed here.

Lines 62-62: "50 Circular Jupiter and Saturn (CJS) (ref. 22), 50 Eccentric Jupiter and Saturn (EJS) (ref. 22), and 141 Grand Tack (ref. 23) simulations." I realize that the authors are experts on N-body simulations for

planetary accretion, but the reader might not be. Hence, it would be good to elaborate here and explain in a few sentences what these models have in common and what sets them apart. This is important because the N-body simulations constitute a key part of the paper.

Line 71: "both have pre-late veneer $\epsilon_{182W} = 2.2 \pm 0.15$ ". Where does the ± 0.15 ϵ -unit uncertainty come from? I believe that the authors should more clearly distinguish between the uncertainty of the W isotope ratio measurements vs. the derived (calculated) pre-late veneer composition of the Earth. Note that both Touboul et al. (2015, Nature) and Kruijer et al. (2015, Nature) report ϵ_{182W} compositions for the Moon that are considerably more precise (on the order of ± 0.05 , 95% conf.). Hence, the $182W$ composition of the Moon is much better known than suggested by the large uncertainty reported in the present version of the manuscript. This should be clarified and/or corrected in a revised version.

Line 73: "mantle WO3" To improve readability maybe point out more explicitly that this is the portion of Earth's W budget hosted in silicates.

Line 81: To make clear that this sentence provides the explanation for the statement made in the previous sentence, maybe add something like "This is because" before "Reduced conditions..."

Line 95: "Theia often has a higher ϵ_{182W} than Earth." Throughout its evolution, or just prior to the giant impact?

Line 101: "then a Moon was formed from Theia material". From this sentence I gather that the Moon is assumed to solely consist of Theia and would therefore not include material from the proto-Earth's mantle whatsoever. However, to my understanding, canonical giant impact scenarios predict that most, but possibly not all of the Moon consists of Theia. Hence why is the terrestrial mantle assumed to not be involved in the mixing during the giant impact? What is the justification for making this assumption? This should be explained more clearly.

Line 115: "time of the Moon forming collision". I presume that this denotes the time after Solar System formation?

Line 130: "With a tolerance of ± 0.15 epsilon units based on measurements". See my comment above. The precision of the isotopic measurements is a factor 2-3 better than suggested in the text. Even though the uncertainty on Earth's inferred (calculated) pre-late veneer ϵ_{182W} value is somewhat larger, this is not equivalent to measurement precision. This should be clarified.

Line 133: "This probability applies to CJS, EJS, and Grand Tack scenarios, and to different values of k and lunar DW". Why are the curves for the Grand Tack model runs slightly offset to those obtained for the CJS/EJS model runs?

Line 136: "a low probability event". Maybe point out that the actual probability(%) values reported in this prior study in fact are very similar to those inferred here using a much more sophisticated approach.

Lines 163-168: "Moon analogues with a Moon-like $f(Hf/W)$ have MWSMA much more similar to Mars than to Earth (Supplementary Fig. S1)" and "Moon analogues with an Earth-like MWSMA have a very high fHf/W of ~ 100 (Supplementary Fig. S1) and are thus unlikely to develop an Earth-like ϵ_{182W} ."

This claim made here is currently not very clearly supported by the model results shown in Fig. S1. Although there are indeed a number of simulations with Mars-like MWSMA reproducing the Moon-like fHf/W , the

$\epsilon^{182}\text{W}$ values are highly variable for a Mars-like MWSMA (~1.5 AU). Moreover, essentially no Theia analogues with Earth-like MWSMA (1 AU) are plotted in the figure. Is there a better way of showing this result and make it more obvious to the reader?

Line 165-166: "so their stable isotopic compositions are expected to be more similar to Mars than to Earth." It is not quite clear what the authors mean here, and some more explanation is required. First, the authors should point out that Mars and Earth have measurably different O isotope compositions. Hence, if Theia derived from a Mars-like MWSMA, as suggested by the model results, then it is expected that the Moon (presuming it derived from Theia) has an O isotope composition that is distinct from the Earth. This is contrary to what is observed.

In addition, have the authors considered the effects of the possible effects of the late veneer on the O isotope compositions of the Earth (see e.g. Herwartz et al., 2014, Science)?

Line 182-183: "Alternative models such as enhanced Earth–Theia mixing (e.g., 7–9) or post-impact equilibration (e.g., 10–11) are likely to have a higher probability of reproducing the observed isotopic compositions."

Given that the main aim of the study is to assess the likelihood of giant impact scenarios, this proposition almost screams for a more quantitative answer. I realize that this is by no means straightforward and probably constitutes a whole new study, but I encourage the authors to at least speculate about the likelihood of alternative models. In addition, some discussion about the relative likelihood of these scenarios from a dynamical viewpoint could be added. Isn't the canonical scenario considered the most likely scenario from a dynamical viewpoint (excluding for a moment caveat of not being able to explain the isotopic similarity)?

Line 213-214: "No terrestrial material was incorporated into the Moon analogue, as an end-member case." See also my comment above. What is the justification for making this assumption, and is this consistent with the canonical giant impact model?

Fig. 1: "Vertical solid lines represent the Moon-forming impact" caption, line 93. The figure panels show multiple vertical solid lines and from the current version it is unclear which of them represents the Moon-forming impact. I presume that it is the line corresponding to the youngest age, but this is not mentioned. I recommend illustrating the time of the Moon forming impact more clearly in the revised version. Also, does the timeline on the horizontal axis denote time after solar system formation? This should be clarified too.

Fig. S1: The assumed lunar DW value of 250 seems very high to me. Why is the justification for using this high W partition coefficient for the Moon? Also, why are only CJS/EJS simulations shown and not those for a Grand Tack scenario?

--

Thomas Kruijer

June 11, 2020

Reviewer #3 (Remarks to the Author):

This is a timely paper that addresses the key outstanding constraint for lunar formation models: how to best explain the isotopic similarities of the Earth and Moon in the context of a giant impact origin. This paper addresses the inferred similar initial Earth-Moon W compositions, perhaps the most challenging isotopic similarity to explain. It concludes that, based on N-body planet accretion models, the likelihood of a Theia and Earth having the needed W isotopic and Hf/W compositions to result in the Earth-Moon in a canonical impact (in which the Moon's composition essentially equals that of Theia) is extremely low. This is a broadly important result. The paper and its methods are generally clearly presented. I recommend publication with some suggested revisions.

p. 2, line 23. 0.08 to 0.38% is quite precise. More on this below.

p. 2, line 29. "the effects of disproportionate late accretion" → "the effects of disproportionate chondritic late accretion" (the quoted equal initial Earth-Moon compositions depend on the chondritic assumption)

p. 3, line 43. For clarity for non-specialists, suggest "A body's W isotopic composition" → "A body's mantle W isotopic composition", so it is clear that the core and mantle will have different compositions within the same body.

p. 4, lines 61-63. A note that these various models reflect different hypothesized giant planet migration histories and timescales would be helpful for the non-specialist.

p. 4, line 65 "continuous" core formation evokes (to this reviewer) smooth exponential functions. But here the core formation is affected by each discrete impact. Consider "ongoing core formation" perhaps.

p. 4, line 71 "both have pre-late veneer" → "both have inferred pre-late veneer"

p. 4, line 72. Please clarify why the W isotopic anomaly would depend on the absolute abundance ratio Hf/W

p. 4, line 75. "inner disk" → "inner protostellar disk", to clarify which disk is being discussed (there is the pre-Moon disk too)

p. 5: Somewhere, probably best in the supplementary information, the authors should clarify how they are defining/identifying their "Earth" and "Theia" analogs. There are several different reasonable ways to do this, but they can yield different results (at least they have for O isotope studies).

p. 7, line 95. "Theia typically grows faster than the Earth" → "Theia typically finishes its accretion earlier" I think this is what is meant, unless a faster growth mode (pebble accretion?) is envisioned for Theia relative to the Earth.

p. 7, line 101 Please comment on how the results would change if modest (up to 20%) contributions from the Earth's mantle to the Moon are considered.

p. 11, line 156. Please comment on why the 5 to 8% O match probability has been considered, rather than the higher probabilities found by Mastrobuono-Battisti and colleagues. As these probabilities are generally quite uncertain, consider fewer significant figures in the final percentages in line 158.

p. 11, lines 165-168. This is a good point that you need Theia to have incorporated oxidized material from the outer disk. So, this argues strongly against getting the "match" in other stable isotopes between Earth and the

Moon by having Theia form in the same radial “zone” as the Earth. But, we know there is some material beyond Mars that (somehow) ended up looking isotopically like the Earth: the ECs. Thus there are things about the initial stable isotope distribution of disk material that we don’t fully understand. The authors do not have to solve this problem! But the issue/uncertainty should be acknowledged.

p. 14, line 214. Again, what is the magnitude of the effect of including a minority Earth contribution in the Moon? It will not substantially change the overall results I think.

p. 14, line 217-220. The varied late veneer description is a little vague. Please clarify how this was done (e.g., was it set to 0.005 Earth masses as is often derived from HSEs, or was it based on the N-body simulations of all the post-Theia accreted material?).

Reviewer #1

I have read and reviewed the manuscript by Fischer, Zube, and Nimmo. This work uses N-body simulations of terrestrial planet formation combined with a geochemical model of core formation to track the evolution of tungsten anomalies within Earth and Theia analogues. These results are then used to show that under several different terrestrial planet formation scenarios, there is only a ~5% or less chance of replicating the Earth's and Moon's similar tungsten anomalies. This result is critical because tungsten anomaly is largely set by accretion timescale and core formation. Meanwhile, other isotopes in which the Earth and Moon are similar are set through the source region of the planetesimal building blocks of Earth and Theia. Because previous works have shown that replicating the Earth and Moon's other isotopic similarities is also ~5%, this means that replicating both tungsten and non-tungsten similarities is exceptionally improbable within the canonical giant impact hypothesis that builds the Moon primarily from Theia material. This implies that either greater mixing took place between the proto-Earth and Theia and/or alternative impact scenarios must be invoked to explain the Earth and Moon's isotopic similarities. The tension between the canonical giant impact hypothesis (in which the Moon is largely derived from Theia material) and the isotopic similarities between Earth and Moon has been the subject of many past papers. This current work is thus highly significant and represents a breakthrough in this topic. After some modest revision, I believe it is suitable for publication.

My limited comments are as follows:

- Much more detail needs to be presented in how the authors chose relevant simulated systems and planets for analysis. How are Earth and Theia analogs defined within N-body simulations? What are the mass and orbital element bounds of Earth analogs? Are Theia analogs the last major impactor on Earth analogs? If so, what is considered to be a major impactor? Moreover, what fraction systems that contain an Earth and Theia analog also have a similar broader architecture to the inner solar system? If this fraction is low, how does this affect the validity of the authors' results?

We have added a few sentences to the Methods section to provide further detail: “The full model identified Earth analogues as the largest surviving planet with a mass of 0.67–1.5 M_{\oplus} and a semimajor axis of 0.75–1.25 AU, and the fast model identified Earth analogues as any surviving planet with a mass of 0.5–2 M_{\oplus} and a semimajor axis of 0.387–1.524 AU. In both models, Theia analogues were defined as the last body to collide with the Earth that contained at least one planetary embryo” (L295–299).

We have also added a reference to the Methods section in the main text: “We tracked the isotopic evolution of Earth and Theia analogues formed in these simulations (Methods) by...” (L83–84).

Quantifying the similarity between a simulation and the Solar System can produce variable results, depending on which and how many aspects are being compared. In general, any given N-body simulation has a low probability of reproducing all aspects of the Solar System. There is some variability depending on accretion

scenario; for example, the Grand Tack has a higher probability of reproducing the observed low mass of Mars, but it also forms planets extremely quickly. We have previously demonstrated that different aspects of the Solar System are uncorrelated in their likelihood of being reproduced (Fischer and Ciesla, 2014, EPSL), and these dynamics should also be uncorrelated with isotopic outcomes (except for their influences on timescales). Therefore, if a simulation produces a Mars analogue that is too large, for example, this should not affect the Moon's W anomaly. We do not focus on how many of these simulations reproduce the Solar System because this has been analyzed in many previous studies (e.g., Fischer and Ciesla, 2014, EPSL; O'Brien et al., 2014, Icarus; Raymond et al., 2009, Icarus; Wetherill, 1994, GCA).

- In the manuscript, the authors consider Moons that are 100% derived from Theia. How much do the results change if the authors make a less stringent requirement? For instance, does replicating the Earth-Moon tungsten anomaly become substantially more plausible if we allow 10% of the Moon to be derived from the proto-Earth? Even more useful for follow-up work studying the giant impact details: how much proto-Earth contribution to the Moon is necessary to increase the replication probability to the tens of percent level?

We thank the reviewers for this great suggestion. In response to comments from all three reviewers, we have performed a new set of calculations in which the Moon is derived from a mixture of Earth and Theia materials, containing up to 50% Earth materials.

We have modified the abstract to reflect this: "We find the probability of forming a Moon from Theia with an Earth-like tungsten isotopic composition is <1.6–4.7%. Mixing in up to 30% terrestrial materials increases this probability, but it remains <10%" (L19–21).

We have clarified in the introduction: "We also consider the formation of the Moon from an Earth–Theia mixture..." (L68).

The section "Terrestrial and lunar ^{182}W anomalies" was also modified: "...with single-stage core formation and no post-impact mixing or equilibration (Methods). As an endmember case, we begin by considering the formation of the Moon from entirely Theia materials, then later discuss the effects of mixing Earth and Theia materials to form the Moon. Fig. 2 shows terrestrial and lunar $\epsilon_{182\text{W}}$... assuming the Moon formed entirely from Theia materials" (L126–131).

We have added a new panel to Figure 3, showing the cumulative probability of various Earth–Moon W anomaly difference as the proportion of Earth materials in the Moon is varied as 0–50% (L179):

and the caption of Figure 3 was expanded to discuss this new panel: “Left panel: Results for four combinations of accretion scenarios and lunar D_W metal–silicate partition coefficient D_W , with the Moon made entirely of Theia materials. In this case, <1.6–4.7% of Earth–Moon pairs have ϵ_{182W} differences of <0.15 ϵ units. Right panel: The effect of mixing Earth and Theia materials to form the Moon, with 0–50% of the mass of the Moon coming from the Earth. Results are shown for CJS/EJS simulations and $D_W = 30$. The probability of an Earth–Moon pair having an ϵ_{182W} difference of <0.15 ϵ units ranges from 1.9% (0% Earth materials) to 9.4% (20–30% Earth materials) to 17.0% (50% Earth materials)” (L186–193).

We added a new paragraph to section “Probability of an Earth-like lunar ϵ_{182W} ” discussing these new results: “This probability can be increased by considering a Moon formed from a mixture of Earth and Theia. For example, for CJS/EJS simulations and a lunar $D_W = 30$, Moons formed entirely of Theia materials have a 1.9% chance of matching the Earth’s ϵ_{182W} within $\pm 0.15 \epsilon$ units. Forming the Moon out of 90% Theia + 10% Earth increases this probability to 3.8%, while using 70–80% Theia + 20–30% Earth increases the probability to 9.4%, and using 50% Theia + 50% Earth results in a 17.0% probability of a match (Fig. 3, right panel). Similar results were found using $D_W = 150$: for example, a Moon formed from 70% Theia + 30% Earth has a 7.5% probability of matching the Earth’s ϵ_{182W} within $\pm 0.15 \epsilon$ units. These probabilities are slightly higher than those reported in a previous study based on a Monte Carlo model (17), which found an Earth–Moon ϵ_{182W} match in <1% of cases when >80% of lunar material derived from Theia, and in <5% of cases when <20% of the Moon derived from Theia, though the results of these two studies are in qualitative agreement that an Earth–Moon match is a low probability event.” (L195–208).

We also modified the Methods section: “For most of the results shown here, no terrestrial material was incorporated into the Moon analogue, as an end-member case. We also considered cases where the lunar mantle was derived from a mixture of Theia and Earth materials, with up to 50% Earth materials by mass” (L310–313).

- The EJS and CJS simulation results are grouped together throughout the paper. Are their results statistically indistinguishable? If not, why are they grouped this way?

We have added a sentence to the Methods section to clarify: “Results from CJS and EJS simulations were combined, due to their resulting isotopic signatures being statistically indistinguishable” (L301–302).

CJS and EJS simulations are similar in many ways and both are quite different from the Grand Tack. In one of our earlier studies (Fischer, Nimmo, and O’Brien, 2018, EPSL), we showed that CJS and EJS simulations typically use similar initial mass distributions and have similar resulting planetary masses and extents of radial mixing, all different from Grand Tack. Additionally, our W isotopic calculations in this study show a high degree of similarity between CJS and EJS (and both different from Grand Tack). For example, for $k=0.4$, Earth analogues in CJS and EJS simulations have median final W anomalies of 1.7 (68% confidence interval of 0.8–4.1) and 2.5 (1.0–6.6), respectively, while the Grand Tack Earth analogues have a higher median anomaly of 4.3. Moon analogues in CJS, EJS, and Grand Tack have median anomalies of 3.8, 4.0, and 7.5, respectively (for $D_W = 30$).

- What is the rationale for choosing $D_W=30$ and $D_W=150$ for the analysis? Can better W anomalies be attained with other values of D_W ? If so, are they physically plausible? I would ask the same question for k .

We tested various values of D_W in the range 30–250. Within this range, we did not find any values of D_W that were systematically more or less successful at producing an Earth–Moon match than other values (though there is some stochastic variation in the probability of a match with different D_W). We have clarified in the text: “values of D_W in the range 30–250 were not found to have significantly different probabilities of an Earth–Moon match” (L167–168).

Previous studies of the Moon’s W isotopic composition have used similar D_W . For example, Kruijer and Kleine (2017) used values of 5–100, Nimmo et al. (2010) used values of 30–150, and Dauphas et al. (2014) used 50–100. This range of D_W is plausible based on experimental measurements of the metal–silicate partitioning of W at Moon-relevant conditions. For example, using the parameterization from Cottrell et al. (2009, 2010, EPSL) for 0.5–2 GPa by weight, for $P = 2$ GPa, $T = 2200$ K, $NBO/T = 2.4$, a D_W of 150 corresponds to an oxygen fugacity of $IW-2.1$, while a D_W of 30 corresponds to $IW-1.5$. There is certainly some uncertainty in the oxygen fugacity of lunar core formation, but this is a reasonable approximate range. We have added references to these studies in the Methods section: “...using a constant D_W in the range 30–250 (e.g., 13, 17, 19, 42–43)” (L314–315).

For both EJS/CJS and Grand Tack calculations, we used values of k ranging from 0.1 (minimal core equilibration) to 1 (complete core equilibration) (e.g., Table S1), given the uncertainty in this value. In calculating the probability of an Earth–Moon match (given a match to the Earth), we combined results from all values of k to achieve better statistics. We have clarified this in the text (“We use values of k ranging from 0.1 to 1, and combine results from different k to achieve better statistics”, L157–158) and in the Figure 3 caption (“Results for various k (in the range 0.1–1) are combined”, L185–186).

- Page 11, lines 171-175: The authors discuss the number of Theia analogues with feeding zones similar to Earth in the CJS and EJS batches and proceed to break down their W statistics. The same should be done for the Grand Tack simulations and their

Theia analogs.

We have added a parallel discussion here of the results of our Grand Tack simulations: “Our CJS/EJS simulations produce 20 Theia analogues with MWSMA < 1.5 AU (similar to the Earth, to match the stable isotope constraints). Of these Theia analogues, 0/20 have $f^{Hf/W} = 8\text{--}30$ (median 1100) and 3/20 have $\epsilon_{182W} = 3.5\text{--}7$ (median 34). Similarly, our Grand Tack simulations produce 186 Theia analogues with MWSMA < 1.5 AU, of which 2/186 have $f^{Hf/W} = 8\text{--}30$ (median 1700) and 14/186 have $\epsilon_{182W} = 3.5\text{--}7$ (median 25), due to the low pressures and temperatures of their interiors, reduced conditions in the inner disk, and rapid accretion” (L238–244).

This concludes my comments.

Reviewer #2

Review of “The origin of the Moon’s Earth-like tungsten isotopic composition from dynamical and geochemical modeling” by Rebecca A. Fischer, Nicholas G. Zube, and Francis Nimmo submitted to Nature Communications.

The study by Fischer et al. aims to shed light on the origin of the Moon by a giant impact, a long-standing debate in planetary science. The key conundrum in this debate is that the Earth and the Moon are isotopically very similar, whereas canonical giant impact scenarios predict a larger fraction of the impactor (Theia) in the Moon, and hence an isotopic difference. The authors aim to tackle this question by assessing the likelihood of producing similar tungsten (W) isotope composition for the Earth and the Moon. To this end, they use sophisticated N-body model simulations for planetary accretion and simultaneously track the W isotopic evolution. The authors find that the probability of producing similar W isotopic compositions in canonical giant impact models is very small (<5%), and even smaller (<1%) if the similar O isotopic compositions of the Earth and Moon are taken into account as well. These results confirm conclusions drawn in other recent work. However, the model calculations used by the authors are considerably more elaborate, allowing for a more rigorous and quantitative assessment of the likelihood of the canonical giant impact scenario. In addition, a novel aspect of this study is that it simultaneously assesses the likelihood of producing similar W and O isotopic compositions for the Earth and the Moon.

The model calculations are state-of-the-art, and as far as I can tell, of excellent quality. The results of these calculations provide new and important insights into the likelihood of lunar formation scenarios, and hence, more generally into the mechanisms of planetary accretion. All things considered, I believe that this is a very timely and important contribution.

However, I do have several comments and suggestions that should be addressed

before the paper can be published. Although the paper generally is very well-written, I believe that some improvements could be made with respect to readability of the paper. Hence, several of my comments below are intended to clarify certain aspects that seemed unclear and/not well enough explained in the current version of the manuscript.

Comments and suggestions (Line by line, most important comments in bold):

Line 14-15: "...coincidentally similar isotopic compositions for the proto-Earth and Theia. Here we evaluate the probability of this third option..." From this I gather that the authors assess the likelihood of having indistinguishable isotopic compositions only in the context of the canonical giant impact scenario. This should be pointed out more explicitly. In addition, it would be good to clarify why the likelihood of alternative giant impact scenarios (e.g. those involving larger contributions of the Earth's mantle and/or post giant impact equilibration) is not assessed in this study.

We have clarified in the abstract "in the context of the canonical giant impact scenario" (L17–18).

We also added a sentence at the end of the introduction: "We also consider the formation of the Moon from an Earth–Theia mixture, but do not directly quantify the probabilities of large extents of mixing or post-impact equilibration since these hypotheses are sensitive to different processes (e.g., dynamics of the Moon-forming impact instead of core formation)" (L68–71).

Lines 41-42: "¹⁸²Hf decays to ¹⁸²W with a half-life of 9 Ma." Yes, but this decay system was only effective early in Solar System history, when ¹⁸²Hf was alive. This should be clarified to avoid confusion.

We have rewritten this sentence to read: "In the early Solar System, the now-extinct ¹⁸²Hf decayed to ¹⁸²W with a half-life of 9 Ma" (L44–45).

Lines 51-53: It would be helpful for the reader to explain how the N-body simulations performed by Nimmo et al. 2010 are different from those performed here.

We have added some more explanation here: "The N-body simulations used in that study (23) are similar to the Circular Jupiter and Saturn (CJS) and Eccentric Jupiter and Saturn (EJS) simulations used here (24), with the main differences being a higher resolution (more initial bodies) in the simulations used here and a much larger number of simulations (100 CJS/EJS in this study versus 8 in ref. 19) to allow for more quantitative assessment of probabilities" (L57–62).

Lines 62-62: "50 Circular Jupiter and Saturn (CJS) (ref. 22), 50 Eccentric Jupiter and Saturn (EJS) (ref. 22), and 141 Grand Tack (ref. 23) simulations." I realize that the authors are experts on N-body simulations for planetary accretion, but the reader might not be. Hence, it would be good to elaborate here and explain in a few sentences what these models have in common and what sets them apart. This is important because the N-body simulations constitute a key part of the paper.

We have added a few sentences of elaboration here: "Each simulation began with ~80–100 larger planetary embryos and ~2000 smaller planetesimals, whose orbital evolutions were tracked for 150–200 Ma until they accreted into planets (ref. 24–25)

(Methods). The simulations also included the Sun, Jupiter, and Saturn. The three accretion scenarios considered here differ primarily in the orbits and migration histories/timescales of the gas giant planets, and their resulting effects on the extent and timescales of radial mixing in the disk (e.g., 26). We tracked the isotopic evolution of Earth and Theia analogues formed in these simulations (Methods) by ...” (L76–84).

We have also added a paragraph to the Methods section to explain in more detail: “The 50 CJS and 50 EJS simulations (24) were run using the MERCURY code (32). The CJS case had Jupiter and Saturn on non-eccentric orbits as predicted by the Nice Model (e.g., 33). In the EJS scenario, Jupiter and Saturn were placed on their modern-day orbits. The CJS/EJS simulations started with ~80 planetary embryos with masses of 0.01–0.06 Earth masses (M_{\oplus}) and ~2000 planetesimals with masses of ~0.001 M_{\oplus} . The 141 Grand Tack simulations (25) were run using the Symba code (34). The Grand Tack scenario involves an inward then outward migration of Jupiter and Saturn to truncate the inner disk (35). These simulations started with ~100 planetary embryos and ~2000 planetesimals. Different simulations featured different embryo:planetesimal mass ratios (1:1, 2:1, 4:1, or 8:1) and different embryo masses (0.025, 0.05, or 0.08 M_{\oplus}). In all CJS, EJS, and Grand Tack simulations used here, all collisions were treated as inelastic mergers, neglecting the possibility of hit-and-run or erosive impacts (e.g., 36). This is a reasonable approximation, as incomplete accretion has been shown to have only a small effect on the resulting ϵ_{182W} (37). These simulations have been previously published, and further details may be found in those studies (24–25)” (L263–278).

Line 71: “both have pre-late veneer $\epsilon_{182W} = 2.2 \pm 0.15$ ”. Where does the ± 0.15 ϵ -unit uncertainty come from? I believe that the authors should more clearly distinguish between the uncertainty of the W isotope ratio measurements vs. the derived (calculated) pre-late veneer composition of the Earth. Note that both Touboul et al. (2015, Nature) and Kruijer et al. (2015, Nature) report ϵ_{182W} compositions for the Moon that are considerably more precise (on the order of ± 0.05 , 95% conf.). Hence, the $182W$ composition of the Moon is much better known than suggested by the large uncertainty reported in the present version of the manuscript. This should be clarified and/or corrected in a revised version.

We are using the ± 0.15 value based on the calculated pre-late veneer composition of the Earth, and because it is more conservative. We have clarified this throughout the text.

Here we have rewritten the sentence to read: “The Earth and Moon both have inferred pre-late veneer $\epsilon_{182W} = 2.2$ (ref. 5), with a measurement uncertainty of approximately ± 0.05 ϵ units on lunar samples (ref. 5–6) and an uncertainty of ± 0.15 ϵ units based on the calculated pre-late veneer composition of the Earth (ref. 5)” (L90–93).

In the Probabilities section, we have modified/added text to clarify: “With a tolerance of ± 0.15 ϵ units based on the uncertainty in the Earth’s pre-late veneer composition (5) ... Using a smaller tolerance than ± 0.15 , more similar to measurement precision (e.g., 5–6), would result in an even lower probability of forming a Moon with an Earth-like ϵ_{182W} ” (L161–166).

We have similarly modified the Figure 3 caption to read: “Vertical dotted lines indicate the maximum Earth–Moon ϵ_{182W} difference allowed based on calculations of the Earth’s pre-late veneer composition, $\pm 0.15 \epsilon$ units (ref. 5)...” (L182–183).

Line 73: "mantle WO3" To improve readability maybe point out more explicitly that this is the portion of Earth's W budget hosted in silicates.

We have added a parenthetical note here: “(the portion of a body’s W budget hosted in silicates)” (L97–98).

Line 81: To make clear that this sentence provides the explanation for the statement made in the previous sentence, maybe add something like “This is because” before “Reduced conditions...”

We have added “This is because...” to the beginning of this sentence (L106).

Line 95: “Theia often has a higher ϵ_{182W} than Earth.” Throughout its evolution, or just prior to the giant impact?

We have added clarification here: “...throughout much of its evolution” (L120).

Line 101: "then a Moon was formed from Theia material". From this sentence I gather that the Moon is assumed to solely consist of Theia and would therefore not include material from the proto-Earth's mantle whatsoever. However, to my understanding, canonical giant impact scenarios predict that most, but possibly not all of the Moon consists of Theia. Hence why is the terrestrial mantle assumed to not be involved in the mixing during the giant impact? What is the justification for making this assumption? This should be explained more clearly.

Thanks for the question. We have added some calculations with incorporation of Earth materials into the Moon. Please see response to Reviewer #1 comment above for all of the changes made, including a clarification here (now L125–129).

Line 115: “time of the Moon forming collision”. I presume that this denotes the time after Solar System formation?

We have added clarification to the caption here: “(expressed as time since Solar System formation)” (L144–145).

In detail, we have calculated these as times since the initialization of the Hf–W isotopic system. For CJS and EJS cases, this is equated with the start of the N -body simulations. A small change in the start time to allow for the formation of planetesimals and planetary embryos before the start of the simulation does not have a noticeable effect on the resulting W anomalies. For the Grand Tack case, 2.4 Ma was added to the simulation run time to account for the formation of these bodies and the migration of Jupiter and Saturn to occur (Zube et al., 2019, EPSL).

Line 130: “With a tolerance of ± 0.15 epsilon units based on measurements”. See my comment above. The precision of the isotopic measurements is a factor 2-3 better than suggested in the text. Even though the uncertainty on Earth's inferred (calculated) pre-late veneer ϵ_{182W} value is somewhat larger, this is not equivalent to measurement precision. This should be clarified.

Please see response to comment above.

Line 133: "This probability applies to CJS, EJS, and Grand Tack scenarios, and to different values of k and lunar DW ". Why are the curves for the Grand Tack model runs slightly offset to those obtained for the CJS/EJS model runs?

We have added a brief discussion of this to the text: "Note that at larger Earth–Moon tungsten anomaly differences, the Grand Tack scenario generally produces lower cumulative probabilities, since its planets form faster, resulting in larger ϵ_{182W} (28)" (L169–171). It is not due to a difference between the codes we used, since we benchmarked them on Grand Tack simulations and got approximately the same answer (Fig. S2).

Line 136: "a low probability event". Maybe point out that the actual probability (%) values reported in this prior study in fact are very similar to those inferred here using a much more sophisticated approach.

We have modified this sentence to read: "Our findings agree well with an earlier study suggesting that this is a low probability event, with a likelihood on the order of 1% (ref. 17)..." (L173–174).

Lines 163-168: "Moon analogues with a Moon-like $f(Hf/W)$ have MWSMA much more similar to Mars than to Earth (Supplementary Fig. S1)" and "Moon analogues with an Earth-like MWSMA have a very high fHf/W of ~ 100 (Supplementary Fig. S1) and are thus unlikely to develop an Earth-like ϵ_{182W} ." This claim made here is currently not very clearly supported by the model results shown in Fig. S1. Although there are indeed a number of simulations with Mars-like MWSMA reproducing the Moon-like fHf/W , the ϵ_{182W} values are highly variable for a Mars-like MWSMA (~ 1.5 AU). Moreover, essentially no Theia analogues with Earth-like MWSMA (1 AU) are plotted in the figure. Is there a better way of showing this result and make it more obvious to the reader?

The previous version of Fig. S1 had its vertical axis truncated, so that many Moon analogues were off scale. We have fixed this in the revised version (and changed the vertical axis to a log scale). We have also added three more panels to this figure, showing the variations in ϵ_{182W} with MWSMA for Earth, Moon, and Mars analogues, and showing all results for both CJS/EJS and Grand Tack, and revised the caption of this figure accordingly.

We have also modified the discussion of this figure in the main text: "Moon analogues with a Moon-like $f^{Hf/W}$ and ϵ_{182W} have MWSMA much more similar to Mars than to Earth (Supplementary Fig. S1). Mars and Earth have different O isotopic compositions (e.g., 31). If Theia originated from a Mars-like MWSMA and the Moon derived primarily from Theia, the Moon would be expected to have a stable isotopic composition more similar to Mars than to Earth, contrary to observations, though there is considerable uncertainty in the initial stable isotope distribution in the protoplanetary disk (e.g., 14). Moon analogues with an Earth-like MWSMA have a very high $f^{Hf/W}$ of ~ 100 and ϵ_{182W} values that are significantly higher than the Moon's on average (Supplementary Fig. S1)" (L226–235).

Line 165-166: "so their stable isotopic compositions are expected to be more similar to Mars than to Earth." It is not quite clear what the authors mean here, and some more explanation is required. First, the authors should point out that Mars and Earth have measurably different O isotope compositions. Hence, if Theia derived from a Mars-like MWSMA, as suggested by the model results, then it is expected that the Moon (presuming it derived from Theia) has an O isotope composition that is distinct from the Earth. This is contrary to what is observed. In addition, have the authors considered the effects of the possible effects of the late veneer on the O isotope compositions of the Earth (see e.g. Herwartz et al., 2014, Science)?

We have expanded this discussion to provide additional clarification; please see response to the previous comment above.

Herwartz et al. (2014) reported a small difference between the oxygen isotopic compositions of the Earth and Moon, which continues to be debated. One of the possible explanations they offer for this difference is the effect of the late veneer on the Earth's composition. We do not attempt to account for this difference, partly because of uncertainties in the initial distribution of oxygen isotopes in the disk, partly because it is still controversial, and most importantly, because the difference they reported between the Earth and Moon is very small (compared to, for example, the difference between the oxygen isotopic compositions of Earth and Mars, given that our most successful Moon analogues have Mars-like MWSMA).

Line 182-183: "Alternative models such as enhanced Earth–Theia mixing (e.g., 7–9) or post-impact equilibration (e.g., 10–11) are likely to have a higher probability of reproducing the observed isotopic compositions."

Given that the main aim of the study is to assess the likelihood of giant impact scenarios, this proposition almost screams for a more quantitative answer. I realize that this is by no means straightforward and probably constitutes a whole new study, but I encourage the authors to at least speculate about the likelihood of alternative models. In addition, some discussion about the relative likelihood of these scenarios from a dynamical viewpoint could be added. Isn't the canonical scenario considered the most likely scenario from a dynamical viewpoint (excluding for a moment caveat of not being able to explain the isotopic similarity)?

We have added some text here discussing the likelihood of other models: "Alternative models such as enhanced Earth–Theia mixing (e.g., 7–9) or post-impact equilibration (e.g., 11–12) are likely to have a higher probability of reproducing the observed isotopic compositions. For example, high energy giant impacts that can potentially allow post-impact isotopic equilibration were common in the end-stages of accretion, with ~85% of bodies experiencing at least one late giant impact with a modified specific impact energy of $>2 \times 10^6$ J/kg (ref. 12). Achieving a high degree of Earth–Theia mixing when the two bodies are similar sizes requires impact velocities that are common in accretion simulations, with ~20% of planets experiencing an impact with a mass ratio of ≥ 0.4 (ref. 7). Future studies should focus on further quantifying the probabilities of these scenarios using dynamical and geochemical constraints" (L251–261).

The canonical scenario may be most likely from a dynamical standpoint, but other scenarios are also possible. More work is needed to quantify these probabilities in future studies.

Line 213-214: "No terrestrial material was incorporated into the Moon analogue, as an end-member case." See also my comment above. What is the justification for making this assumption, and is this consistent with the canonical giant impact model?

Please see response to Reviewer #1 comment above, including changes to the Methods section here (now L310–313).

Fig. 1: "Vertical solid lines represent the Moon-forming impact" caption, line 93. The figure panels show multiple vertical solid lines and from the current version it is unclear which of them represents the Moon-forming impact. I presume that it is the line corresponding to the youngest age, but this is not mentioned. I recommend illustrating the time of the Moon forming impact more clearly in the revised version. Also, does the timeline on the horizontal axis denote time after solar system formation? This should be clarified too.

We have added black arrows to Fig. 1 to more clearly indicate the Moon-forming impacts. We have also modified the figure caption to read: "Thin vertical solid lines with black arrows represent the Moon-forming impacts" (L117–118).

Fig. S1: The assumed lunar D_W value of 250 seems very high to me. Why is the justification for using this high W partition coefficient for the Moon? Also, why are only CJS/EJS simulations shown and not those for a Grand Tack scenario?

Given the uncertainties in oxygen fugacity, temperature, etc. of lunar core formation, a D_W of 250 is within the plausible range based on experimental data (e.g., Cottrell et al., 2009, 2010, EPSL). We use 250 for this figure because it gives a qualitatively good match to the Moon's $f^{Hf/W}$. Even using a significantly lower value of D_W does not produce a Moon-like f at an Earth-like MWSMA; this would require a value of $D_W \ll 1$ (lithophile behavior for W), which is not plausible.

As mentioned in response to a comment above, we have now added new panels to this figure showing the Grand Tack results, which exhibit the same general trends.

--

*Thomas Kruijer
June 11, 2020*

Reviewer #3

This is a timely paper that addresses the key outstanding constraint for lunar formation models: how to best explain the isotopic similarities of the Earth and Moon in the context of a giant impact origin. This paper addresses the inferred similar initial Earth-

Moon W compositions, perhaps the most challenging isotopic similarity to explain. It concludes that, based on N-body planet accretion models, the likelihood of a Theia and Earth having the needed W isotopic and Hf/W compositions to result in the Earth-Moon in a canonical impact (in which the Moon's composition essentially equals that of Theia) is extremely low. This is a broadly important result. The paper and its methods are generally clearly presented. I recommend publication with some suggested revisions.

p. 2, line 23. 0.08 to 0.38% is quite precise. More on this below.

We have rounded these numbers so that they only contain one significant digit, 0.08–0.4% (L26, 219, 249).

p. 2, line 29. “the effects of disproportionate late accretion” ◊ “the effects of disproportionate chondritic late accretion” (the quoted equal initial Earth-Moon compositions depend on the chondritic assumption)

We have added the word “chondritic” here (L32).

p. 3, line 43. For clarity for non-specialists, suggest “A body’s W isotopic composition” ◊ “A body’s mantle W isotopic composition”, so it is clear that the core and mantle will have different compositions within the same body.

We have added the word “mantle” here for clarification (L47).

p. 4, lines 61-63. A note that these various models reflect different hypothesized giant planet migration histories and timescales would be helpful for the non-specialist.

We have added several sentences of explanation here and a new paragraph to the Methods section; please see response to Reviewer #2 comment above.

p. 4, line 65 “continuous” core formation evokes (to this reviewer) smooth exponential functions. But here the core formation is affected by each discrete impact. Consider “ongoing core formation” perhaps.

We have changed the word “continuous” to “protracted” (L84).

p. 4, line 71 “both have pre-late veneer” ◊ “both have inferred pre-late veneer”

We have added the word “inferred” here (L91).

p. 4, line 72. Please clarify why the W isotopic anomaly would depend on the absolute abundance ratio Hf/W

We have added text here to clarify this: “ ϵ_{182W} depends on the abundance of radiogenic ^{182}Hf (and thus Hf in general), as well as the abundance of a stable isotope of W. Therefore it is sensitive to the relative concentrations of Hf and W, quantified as $f...$ ” (L93–96).

p. 4, line 75. “inner disk” ◊ “inner protostellar disk”, to clarify which disk is being discussed (there is the pre-Moon disk too)

We have clarified this as “inner circumstellar disk” (L100).

p. 5: Somewhere, probably best in the supplementary information, the authors should clarify how they are defining/identifying their “Earth” and “Theia” analogs. There are several different reasonable ways to do this, but they can yield different results (at least they have for O isotope studies).

We have made this change, please see response to Reviewer #1 comment above.

p. 7, line 95. “Theia typically grows faster than the Earth” \diamond “Theia typically finishes its accretion earlier” I think this is what is meant, unless a faster growth mode (pebble accretion?) is envisioned for Theia relative to the Earth.

We have modified this phrase to read “Theia typically finishes its accretion earlier than Earth” (L120–121).

p. 7, line 101 Please comment on how the results would change if modest (up to 20%) contributions from the Earth’s mantle to the Moon are considered.

Thanks for the suggestion, we have added this calculation to the paper. Please see response to Reviewer #1 comment above for details.

p. 11, line 156. Please comment on why the 5 to 8% O match probability has been considered, rather than the higher probabilities found by Mastrobuono-Battisti and colleagues. As these probabilities are generally quite uncertain, consider fewer significant figures in the final percentages in line 158.

See response to comment above: we have decreased the number of significant figures on these joint probabilities. We have also added the following sentence to take into account the differing estimates of Mastrobuono-Battisti et al. and Pahlevan and Stevenson: “This probability may be lower by a factor of >3 –4 (ref. 11) or higher by a factor of ~ 4 –5 (ref. 15) if different estimates for the probability of an Earth-like lunar O isotope signature are used” (L219–221). We use the 5–8% probability in the abstract and final paragraph because it is the intermediate of the three estimates.

p. 11, lines 165-168. This is a good point that you need Theia to have incorporated oxidized material from the outer disk. So, this argues strongly against getting the “match” in other stable isotopes between Earth and the Moon by having Theia form in the same radial “zone” as the Earth. But, we know there is some material beyond Mars that (somehow) ended up looking isotopically like the Earth: the ECs. Thus there are things about the initial stable isotope distribution of disk material that we don’t fully understand. The authors do not have to solve this problem! But the issue/uncertainty should be acknowledged.

We have expanded the discussion of this issue (see response to Reviewer #1 comment above), and also added the caveat that “there is considerable uncertainty in the initial stable isotope distribution in the protoplanetary disk (e.g., 14)” (L231–233).

p. 14, line 214. Again, what is the magnitude of the effect of including a minority Earth contribution in the Moon? It will not substantially change the overall results I think.

Please see response to Reviewer #1 comment above.

p. 14, line 217-220. The varied late veneer description is a little vague. Please clarify how this was done (e.g., was it set to 0.005 Earth masses as is often derived from HSEs, or was it based on the N-body simulations of all the post-Theia accreted material?).

We have clarified in the Methods section here that the late veneer was “defined as all planetesimals accreted by an Earth analogue after the Moon-forming impact, regardless of mass” (L318–319).

REVIEWERS' COMMENTS

Reviewer #1 (Remarks to the Author):

To Whom It May Concern:

I have read through the revised manuscript as well as the authors' responses to my comments. I thank them for carefully considering my comments and altering their work in response. I wholeheartedly recommend this paper for publication.

Reviewer #2 (Remarks to the Author):

The authors did an excellent job in revising the manuscript according to the reviewers' suggestions. I do not have any further comments at this stage, and hence recommend publication of the article without further revisions.